# Hydrogenosome, Pairing Anaerobic Fungi and H_2_-Utilizing Microorganisms Based on Metabolic Ties to Facilitate Biomass Utilization

**DOI:** 10.3390/jof8040338

**Published:** 2022-03-24

**Authors:** Jing Ma, Pei Zhong, Yuqi Li, Zhanying Sun, Xiaoni Sun, Min Aung, Lizhuang Hao, Yanfen Cheng, Weiyun Zhu

**Affiliations:** 1Laboratory of Gastrointestinal Microbiology, National Center for International Research on Animal Gut Nutrition, Nanjing Agricultural University, Nanjing 210095, China; 2020105042@stu.njau.edu.cn (J.M.); 14119117@njau.edu.cn (P.Z.); 2018105039@njau.edu.cn (Y.L.); sunzhanying@njau.edu.cn (Z.S.); sunnyxiaoni@njau.edu.cn (X.S.); minaung.uvs@gmail.com (M.A.); zhuweiyun@njau.edu.cn (W.Z.); 2Department of Animal Nutrition, University of Veterinary Science, Nay Pyi Taw 15013, Myanmar; 3Key Laboratory of Plateau Grazing Animal Nutrition and Feed Science of Qinghai Province, State Key Laboratory of Plateau Ecology and Agriculture, Qinghai Plateau Yak Research Center, Qinghai Academy of Science and Veterinary Medicine of Qinghai University, Xining 810016, China; lizhuanghao1122@foxmail.com

**Keywords:** anaerobic fungi, hydrogenosome, fiber utilization, renewable resources

## Abstract

Anaerobic fungi, though low in abundance in rumen, play an important role in the degradation of forage for herbivores. When only anaerobic fungi exist in the fermentation system, the continuous accumulation of metabolites (e.g., hydrogen (H_2_) and formate) generated from their special metabolic organelles—the hydrogenosome—inhibits the enzymatic reactions in the hydrogenosome and reduces the activity of the anaerobic fungi. However, due to interspecific H_2_ transfer, H_2_ produced by the hydrogenosome can be used by other microorganisms to form valued bioproducts. This symbiotic interaction between anaerobic fungi and other microorganisms can be used to improve the nutritional value of animal feeds and produce value-added products that are normally in low concentrations in the fermentation system. Because of the important role in the generation and further utilization of H_2_, the study of the hydrogensome is increasingly becoming an important part of the development of anaerobic fungi as model organisms that can effectively improve the utilization value of roughage. Here, we summarize and discuss the classification and the process of biomass degradation of anaerobic fungi and the metabolism and function of anaerobic fungal hydrogensome, with a focus on the potential role of the hydrogensome in the efficient utilization of biomass.

## 1. Introduction

The rumen is known to be a site for efficient degradation of lignocellulosic biomass due to its well-developed microbial ecosystem. Anaerobic fungi form a unique group among these microorganisms. Although bacteria and protozoa are the dominant microorganisms in most hosts [1], the percentages of anaerobic fungi of the gut microflora in these hosts are relatively small, accounting for about 7–9% [2], they are the important colonizers and degraders for the poorer-quality roughage [3], and the number of anaerobic fungi rises with the increase in dietary fiber [4,5].

The hydrogenosome is a special metabolic organelle in anaerobic fungi. It metabolizes pyruvate and malate obtained through glycolysis and eventually produces ATP and releases organic acids such as formate and acetate, as well as gases such as H_2_ and CO_2_ [6]. The hydrogenosome plays an important role in the growth and metabolism of anaerobic fungi. On the one hand, it produces ATP, an essential energy source for the growth; on the other hand, the accumulation of the hydrogenosomal metabolites (H_2_, formate) inhibits the metabolism in anaerobic fungi [7]. The timely removal of these products leads to the production of NAD (P)^+^ in the hydrogenosome and further subsequent metabolism [8]. In such a case, the metabolism of the hydrogenosome and anaerobic fungi is enhanced, and the degradation of polysaccharide is improved [9]. In particular, when the inhibition of the hydrogenosome is removed, metabolic flow is diverted from the cytoplasm to the hydrogenosome [10].

H_2_ produced by the hydrogenosome could be utilized by some microorganisms to generate other valued organic matter. In agriculture, this beneficial interaction can be used to treat lignocellulosic biomass to produce nutrient components beneficial to ruminants. Additionally, in industry this interaction can help recover energy from low-quality biomass. Therefore, the study of the hydrogenosome is important for the subsequent development of anaerobic fungi into strains that can be industrially used to efficiently utilize crude fibers and for gaining novel insights into the energy resource production.

## 2. An Overview of Anaerobic Fungi

In 1975, Orpin first reported the presence of anaerobic fungi in the rumen of sheep [11]. Since then, many researchers have successively found these microorganisms in the rumen, foregut, and hindgut of many herbivores, suggesting their widespread presence in the digestive tract of herbivores. Anaerobic fungi have been found in almost all animals that ferment in the predigestive tract, including ruminants (e.g., Bovidae, Cervidae), pseudoruminants (e.g., hippos, camels, llamas, alpacas), and nonruminants (e.g., wallabies). They are also found in many postdigestive tract fermenters that digest plant tissues in the cecum and large intestine (e.g., elephants, horses, and rhinoceroses) and in some large herbivore rodents (e.g., long-eared guinea pigs). It appears that the establishment of anaerobic fungi in the alimentary canal of herbivores may be attributed to the complex and distinct chamber with a relatively neutral pH in the digestive tract and a long lag time in the digestive process for the ingested plant tissue [12], which is conducive to the growth and activity of anaerobic fungi. Therefore, anaerobic fungi, which appear in the digestive tract of these herbivores, especially in those that take in a lot of roughage [4,5], must have some unique reasons and advantages to exist in such a complex environment. The study of anaerobic fungi is increasingly becoming an important part of animal nutrition.

### 2.1. Classification of Anaerobic Fungi

Early after the discovery of anaerobic fungi, they were considered to be protozoa with flagella due to their zoospores [11]. In the 1970s, Orpin suggested that these flagellates were actually the transmission stage of fungal zoospores [13], and that chitin components in their cell walls [14] further confirmed their correct location in the fungal kingdom. Since then, a series of related experiments have established the classification system of the lignocellulosic-degrading fungi and the corresponding functional mechanism.

Anaerobic fungi belong to the fungi kingdom, phylum Neocallimastigomycota, order Neocallimastigales, and family Neocallimastigaceae. Up to now, 20 culturable genera of anaerobic fungi have been reported, containing *Neocallimastix* [15], *Caecomyces* [16], *Orpinomyces* [17], *Piromyces* [17], *Anaeromyces* [18], *Cyllamyces* [19], *Buwchfawromyces* [20], *Oontomyces* [21], *Pecoramyces* [22], *Feramyces* [23], *Liebetanzomyces* [24], *Agriosomyces* [25], *Aklioshbomyces* [25], *Capellomyces* [25], *Ghazallomyces* [25], *Joblinomyces* [25], *Khoyollomyces* [25], *Tahromyces* [25], *Aestipascuomyces* [26] and *Paucimyces* [27]. These isolated and cultured genera, however, only occupy a part of the natural anaerobic fungal communities, as revealed by a large and increasing number of sequences in public databases analyzed by a variety of molecular analysis tools [12,28]. At first, the classification of anaerobic fungi mainly depended on their morphological characteristics, such as monoflagellate fungi with less than four flagella on spores and polyflagellate fungi with more flagella, monocentric fungi with nuclei only in sporangia and polycentric fungi with nuclei in both sporangia and rhizoid system, and fungi with filamentous or bulbous rhizoid. However, when the nutritional environment changes, these morphological characteristics of anaerobic fungi also change. At the same time, morphological identification relies on the anaerobic cultivation, which identifies culturable species with a very limited range and gives rise to the great controversy and challenges of the morphological identification of anaerobic fungi. The development of molecular technology has broadened insight into microbial diversity and provided many practical approaches for the isolation and identification of anaerobic fungi.

The 18S rRNA gene sequence analysis technique was first used as a method for the evolutionary classification of anaerobic fungi. Bowman et al. used this technique to determine the location of anaerobic fungi in the class Chytridiomycetes [29]. Then Li and Heath believed that anaerobic fungi should be classified into the order Neocallimasticales, according to the 18S rRNA sequence structure [30]. More and more 18S rRNA gene sequences of anaerobic fungi have been uploaded to the gene database, and these sequences in the small ribosomal subunit (SSU) is highly conserved in Neocallimastigomycota, so its classification at the genus level is not very accurate and has certain limitations [31,32]. The internal transcribed spacer (ITS) is now becoming a commonly used fungal molecular marker, which consists of ITS1 (between 18S rRNA and 5.8S rRNA) and ITS2 (between 5.8S rRNA and 28S rRNA) in rRNA. ITS is relatively stable in the evolutionary process of fungi, and the hypervariable region of ITS has a greater relative change compared with 18S rRNA, so it is more suitable for studying the evolutionary relationship between fungi. ITS1 has been widely used to compare and classify anaerobic fungi of different species [33,34,35]. Although the ITS1 region of anaerobic fungi can be used for environmental sample analysis, the accuracy of this method is affected by the sample community composition. In addition, the fuzziness of sequence annotation in pure culture caused by the heterogeneity of ITS1 reinforces the limitations of the ITS1 region in the classification of anaerobic fungi [36]. Except ITS1, some researchers use the large ribosomal subunit’s (LSU) D1/D2 region as a symbol of the classification of anaerobic fungi, because these areas between different genera and species of fungi have more stable variation [37]. Compared with the method ITS1, LSU alignment for anaerobic fungal classification is more conserved due to the much lower degree of heterogeneity between the sequences and much less frequent presence of large insertions/deletions in some sequences. This makes LSU phylogeny avoid a misleading assignment owing to complicated variation, especially excessive intragenomic variation, which always leads to the ambiguous ITS1 alignment. For specific detection and phylogenetic placement of anaerobic fungi, the LSU method targeting the 28S rRNA gene appears to constitute a consistent and more reliable phylogenetic barcode [38]. The combination of ITS1 and LSU for the identification of anaerobic fungi can compensate for each other’s limitations, and for some species still in dispute, another molecular marker may need to be developed to determine their location.

Although there has been a rapid progress of molecular biology, which has promoted the development of anaerobic fungal genomics, it is a challenge to conduct the sequencing and identification and accurate classification of anaerobic fungi due to the high A-T content (~80%) [2] in their genome. So far, whole-genome sequencing has been published (Table 1) only for 11 species of 20 genera. More suitable molecular techniques should be developed to address these challenges.

### 2.2. Digestion of Plant Fiber by Anaerobic Fungi

There are a lot of lignocellulolytic bacteria in rumen. These bacteria secrete carbohydrate-active enzymes (CAZymes) to help digest complex and recalcitrant lignocellulose material of roughage in the host. In comparison with bacteria, anaerobic fungi not only possess physical penetration effect on plant tissue, but also secret a group of highly specialized CAZymes, including GH10, GH11, GH6, GH45, GH5, GH43, CE1, CE4, and and so on with higher abundance than bacteria during the enzymatic process [44].

#### 2.2.1. Physical Degradation with Fungal Rhizoids

Anaerobic fungi use flagella as their locomotor organ, and the chemotaxis of moving fungal zoospores to soluble sugars enables them to move quickly ahead of other rumen microorganisms and colonize on the ingested plant tissue [45]. It may provide a unique advantage for fewer anaerobic fungi to compete for nutrients in the complex rumen environment [46]. When they touch the surface of the plant, the zoospores spread out their flagella to form cysts. During the formation and growth of cysts, the rhizoids originate from the side of the cell opposite the insertion of the flagella, and is polar or lateral in position [13], and these develop into a highly branched rhizoid system. The rhizoid system of the anaerobic fungi then penetrates the plant tissue and destroys its structure. This first step is physical degradation, which exposes a larger area for interaction between digestive enzymes and fibrous tissue, and releases the plant cell contents for utilization by itself (fungi) and other rumen microorganisms. With the invasion of the rhizoid system, a series of CAZymes are secreted to release cellulose, hemicellulose, and oligosaccharides from lignocellulose and convert these polymers into soluble sugars. These sugars could be used by hosts and microbiota within the system [2,12,47,48]. The important developmental stages of anaerobic fungi and their physical digestion of feedstuff can be clearly observed under a microscope (Figure 1).

#### 2.2.2. Digestion by Diverse Plant Fiber-Degrading Enzymes

The efficient degradation of forage by anaerobic fungi is also attributed to the complete variety of carbohydrate-active enzymes (CAZymes), including glycoside hydrolases (GHs), carbohydrate esterases (CEs), polysaccharide lyases (PLs), glycosyl transferases (GTs). Carbohydrate-binding modules (CBMs) and less studied auxiliary activities (AAs) are accessories that are functionally related to CAZymes. Genome information of some anaerobic fungal strains that have been genetically annotated showed the presence of rich CAZymes, CBMs, and AAs of different types (Figure 2). Details on activities of these enzymes can be found in [44]. The most diverse group of CAZymes in the anaerobic fungi are the GHs [44]. Enzymes hydrolyzing cellulose, hemicellulose, pectin, and other plant wall polysaccharides constitute a large group of GHs. Among these enzymes, cellulases and hemicellulases are the main ones.

Cellulase catalyzes the hydrolysis of β-1, 4-glucosidase in cellulose. Three types of cellulases (endoglucanase, exoglucanase, and β-glucosidase) that have been reported in anaerobic fungi together hydrolyze cellulose and release glucose [51]. Endoglucanase is the most important component of the cellulase system, which can cleave the glucan chain inside the cellulose. High levels of endoglucanase were found in the culture supernatants of *Neocallimastix frontalis* [52,53], *Piromyces communis* [54], and *Orpinomyces* sp. [55]. Endoglucanase production was maximum when the fungi were grown on cellulose, whereas synthesis was totally repressed by the addition of glucose, indicating that the enzyme was subject to regulation [52]. Exoglucanase cleaves cellobiose units, the building blocks of cellulose, from the ends of the cellulose chain. Exoglucanase active against microcrystalline cellulose was also detected, but at lower levels than endoglucanase [52,55]. In addition, β-glucosidase cleaves cellobiose, which is a potent inhibitor of the former two enzymes, to yield glucose [56]. Previous studies have classified that the glycoside hydrolase enzyme classes GH1, GH2, GH3, GH5, GH6, GH8, GH9, GH38, GH45, GH48, and GH74 encode for cellulases [57]. Enzymes in different GHs have a different fiber hydrolysis activity according to their structure. The expression of these classes of enzymes varies according to fungal species and provided growth substrates. GH6 and GH5 were the two highest-expressed families in the *Pecoramyces* sp. F1 according to its transcriptome analysis, whereas GH1 and GH33 were the lowest-expressed ones [58]. In the genome of *Pecoramyces ruminantium* C1A, GH45 and GH48 were highly expressed with glucose as substrate, while the expression of GH6 was dramatically improved by corn stover [59].

Hemicellulose is the main component of the plant primary wall. It is a heteropolysaccharide containing xylan, glucuronoxylan, arabinoxylan, glucomannan, galactomannan, and xyloglucan [60,61]. The component monosaccharides are connected to each other, forming a hard part of the cell wall to prevent microorganisms from using the plant cell contents. The hemicellulases secreted by anaerobic fungi include xylanases, mannanases, galactanases, β-glucanases and so on, which can degrade hemicellulose into various oligomers and monosaccharides. Due to xylan being the basic polymeric compound of hemicellulose and the high hydrolase activity of xylanases, xylanases are the most studied enzymes to date among the hemicellulose hydrolases from anaerobic fungi [62,63,64]. Heterologous expression of these enzymes has also been developed. Xue et al. isolated a *Neocallimastir patricianun* xylanase cDNA and engineered it for heterologous expression in *Escherichia coli* (*E. coli*). The modified xylanase produced in *E. coli* had a specific activity of 1229 U mg^−1^ protein at pH 7 and 50 °C, without purification [65]. The genes encoding xylanases from *Neocallimastix* sp. GMLF2 [66], *Orpinomyces* sp. Strain 2, and *Neocallimastix frontalis* [67] were also cloned into *E. coli*, and a high expression was obtained. In addition to *E. coli*, *Hypocrea* sp. [68], *Kluyveromyces* sp., and *Pichia* sp. [67] may also be ideal heterologous expression vectors for xylanases from anaerobic fungi. Successful development of heterologous production technologies for the enzymes from anaerobic fungi constitutes a significant advancement in applying this promising source of genes towards lignocellulose bioconversion. The glycoside hydrolase classes GH10, GH11, GH30, GH31, GH38, GH39, GH43, GH47, GH53, and GH115 encode for hemicellulases, respectively [57]. Xylanases are representative enzymes of GH10 and GH11. The former has a relatively high molecular weight, while the latter has a lower molecular weight.

In addition to the free enzymes secreted outside the cell, anaerobic fungi also secrete a number of large (MDa) multiprotein cellulolytic complexes, known as cellulosomes, which can degrade crude fibers more completely and efficiently. Although most studies have reported the function of cellulosomes in anaerobic microorganisms, such as anaerobic fungi, and the bacteria *Clostridium thermocellum* and *Ruminococcus albus*, there have also been reports on the discovery of cellulosomes in aerobic bacteria [69]. Cellulosomes include carbohydrate-binding domains, noncatalytic protein domains, and many glycosyl-hydrolases (cellulases, hemicellulases, pectin enzymes, chitinases, and other enzymes). The assembly of cellulosomes involves interactions between dockerin domains on different catalytic enzyme subunits and cohesin domain on noncatalytic scaffolding, and the specific interactions between them allow various enzyme proteins to bind stably in this supramolecular structure. Scaffold proteins and some enzymes contain CBMs, which promote the binding of the cellulosome enzyme to substrate cellulose. Cellulosomes are attached to the cell surface by additional anchoring domains by noncovalent bonding [70]. Many GHs of anaerobic fungi are involved in the formation of cellulosomes. It was reported that anaerobic fungal cellulosomes exhibit additional 13% higher GH activity due to the presence of GH3, GH6, and GH45 compared with bacterial cellulosomes; especially the supplementary β-glucosidase conferred by GH3 activity empowers fungal cellulosomes in converting cellulose to single simple sugars (monosaccharides) when compared with low-molecular-weight oligosaccharide-generating bacterial cellulosomes [39,71]. Because of the powerful fiber degradation capability of cellulosomes, better than commercial preparations containing noncomplexed enzymes, the efficient degradation of cheap fibrous materials is possible if the modified cellulosome genes can be inserted into appropriate host cells and expressed using genetic engineering techniques.

## 3. An Overview of the Hydrogenosome

Hydrogenosome is a kind of membrane-bound organelle that widely exists in some evolutionarily distant protozoa and fungi, such as trichomonas, anaerobic fungi, endoamoeba and microsporidia. These microorganisms are anaerobic or microanaerobic, and they do not have mitochondria. Instead, they rely on the hydrogenosome, a kind of mitochondrion-related organelles (MROs), to metabolize organic matter under anoxic conditions, producing ATP to maintain their metabolism and growth [72,73].

### 3.1. The Origin of the Hydrogenosome

Two billion years ago, the concentration of oxygen (O_2_) in the atmosphere rose sharply because of the photosynthesis of ocean surface cyanobacteria, which changed the proportion of the main components in the atmosphere. The transition from an anaerobic to an aerobic environment had a serious impact on the survival of organisms. In an aerobic environment, some aerobic bacteria (such as *α-proteobacteria*) can utilize organic matter and use O_2_ as an electron acceptor to produce ATP for energy through the process of tricarboxylic acid cycle (TCA) and electron transport. After being swallowed by an amitochondriate eukaryotic host cell or an archaeal cell [74], these ancient aerobic bacteria may have evolved into respiratory organelles, known as mitochondrion, in a long-term symbiotic relationship. Mitochondria are important sites for O_2_ utilization, energy production, material conversion, and metabolic regulation, and it has been thought as the organelles common to eukaryotic cells for a long time. However, since the 1980s, researchers have found that there are some obligatory or facultative anaerobic protozoa and fungi, such as trichomonas, anaerobic fungi, amoeba, and microsporidia that do not contain mitochondria, but a kind of MROs, including hydrogenosome, mitosome, and so forth, which can transfer electrons in other ways and release ATP to supply energy for cells.

The question about the origin of the hydrogenosome has puzzled many scientists. The presence of hydrogenosome in diverse evolutionarily distant organisms (Figure 3) indicates that it has evolved independently on several occasions.

Earlier, no organellar DNA was detected or isolated from the hydrogenosome, which hampered the direct genetic demonstration of its evolutionary ancestry. Until 2005, phylogenetic analysis of the integrant hydrogenosomal gene of *Nyctotherus ovalis* provided explicit evidence that the hydrogenosome is indeed a modified mitochondrion [75]. However, since the enzymes and metabolism progress of the hydrogenosome in different organisms are not exactly the same, and there is no complete series of evidence to elucidate its evolution, different hypotheses about the origin of the hydrogenosome exist. These hypotheses can ultimately be generalized or covered in two arguments, whether the hydrogenosome is a degraded form of mitochondria that lost some functional proteins and gene fragments to adapt to an anaerobic environment, or the two organelles originate from the same or different endosymbionts to cope with different atmospheric conditions and energy driving forces (Figure 4).

For decades, researchers have conducted a large number of experiments using molecular biology, molecular genetics, and systematic taxonomy to find out the origin of the hydrogenosome. There is a large body of evidence, for example, their clearly semblable appearance [76], their closely related enzymes with important functions [77,78,79], and analogous organelle gene sequences [75,80], which support both hypotheses. Nowadays, most researchers who work on the hydrogenosomes agree with the view that these organelles originated from mitochondrion, losing partial or all mitochondrial genome in anaerobic environments.

### 3.2. Structure and Function of the Hydrogenosome

#### 3.2.1. Structure of the Hydrogenosome

Most hydrogenosomes are spherical or slightly elongated granules, with a homogeneous particle matrix inside. However, in some cells, the hydrogenosomes are not spherical but are very elongated, such as those found in *Monocercomonas* sp. [81]. The hydrogenosomes found so far are encased by two adjacent membranes, which are very thin, presenting a thickness of only a few nanometers [82]. Although researchers have reported that the hydrogenosome of *Neocallimastix* sp. L2 is monomembraned [83], Benchimol et al. modified the traditional fixation procedure with CaCl_2_ and, through transmission electron microscopy and frozen sections, found that the inner and outer membranes are very close to each other [82]. This indicated that an appropriate fixation method is necessary for good visualization of the two membranes. In addition, researchers have found that in some regions of the hydrogenosome of *Trichomonas foetus* (*T*. *foetus*), the separation between the two membranes forms a vesicle structure. Under transmission electron microscopy, the electron density of the vesicles was slightly higher than that of the matrix [84]. The number, size, and electron density of the vesicles varied in different microorganisms [11,84,85,86]. These vesicles may be related to the storage of calcium and other cations and change at different metabolic states.

#### 3.2.2. Function of the Hydrogenosome

In 1973, Lindmark and Muller discovered that the organelles in *T. foetus* did not possess the typical mitochondrial characteristics of the tricarboxylic acid cycle, electron transport chain, F_0_–F_1_ ATPase, and oxidative phosphorylation, but the enzyme system in the particle matrix was capable of completing the anaerobic metabolism of pyruvate. In contrast to mitochondria and peroxisomes, these organelles directly use protons as terminal electron receptors to produce H_2_; therefore, they proposed a name for this subcellular particle called hydrogenosome in conjunction with previous studies [85]. The hydrogenosome converts pyruvate into various metabolites, such as formate, acetate, H_2_, and CO_2_, and also produce ATP through substrate-level phosphorylation for cell growth, acting as an oxygen-independent mitochondrion.

In addition, the hydrogenosome possesses another important function for its survival known as antioxidative stress response. The enzymes in the hydrogenosome, especially pyruvate: ferredoxin oxidoreductase (PFO) and hydrogenase, are highly sensitive to O_2_ and can be inactivated rapidly under the condition of O_2_, blocking the substance metabolism in the hydrogenosome. The inhibition of the enzyme by O_2_ may be due to the formation of reactive oxygen species (ROS). For example, O_2_ is thought to bind to [Fe] at the far end of the [2Fe]H subcluster, then form an ROS and destroy the [4Fe4S]H subcluster to inactivate the [FeFe] hydrogenase [87]. In most living organisms, iron-dependent superoxide dismutase (SOD) converts ROS to hydrogen peroxide (H_2_O_2_), which is subsequently removed by mitochondria usually using the glutathione system and catalase. The mechanism by which the hydrogenosome defends against the damage from ROS is not entirely the same. Although Lindmark et al. reported SOD activity in *T*. *foetus* hydrogenosomes [88], there have been no reports of peroxide reductase in the organelles that break down H_2_O_2_. Unlike mitochondria, thioredoxin-linked peroxiredoxin antioxidant system is one of the major antioxidant defense mechanisms in trichomonas hydrogenosomes. Three relevant peroxidases in the hydrogenosome of *Trichomonas vaginalis* (*T*. *vaginalis*), thioredoxin reductase (TrxR), thioredoxin (Trx), and Trx-dependent peroxidases (TrxP) jointly play a role in the hydrogenosome to reduce oxidative stress caused by H_2_O_2_. Moreover, when *T**. vaginalis* is subjected to oxidative stress, the expression of Trx and TrxP is increased at both transcriptional and translational levels, demonstrating that the parasite is capable of responding to exogenous oxygen stress through changes in both levels [89].

### 3.3. Metabolism in the Hydrogenosome

#### 3.3.1. Carbohydrate Metabolism

Due to the difference in structure and enzyme, the hydrogenosomal metabolism in different microorganisms is not exactly the same. In general, there are many enzymes related to carbohydrate metabolism in the hydrogenosome, including malic enzyme (ME), PFO, [2Fe-2S] ferredoxin protein (Fdx), succinate thiokinase (SCS), adenosine kinase (AK), [Fe] hydrogenase (Hyd), acetate: succinyl CoA-transferase (ASCT), among others. Glucose enters the glycolysis process when it is transferred into the cytoplasm by glucose transporters. Then the formed malate or pyruvate goes into the hydrogenosome; it is metabolized as CO_2_, formate, and acetate and finally releases energy in the form of ATP. PFO and hydrogenase are two hydrogenosomal enzymes that do not exist in mitochondria. Instead of the pyruvate dehydrogenase enzyme complex, PFO converts the pyruvate produced by glycolysis to acetyl CoA and CO_2_ (by taking off the hydroxyl group). Then the consequent electron is passed to the middle electronic carrier Fdx, and the reduced Fdx is oxidized by ferredoxin hydrogenase, eventually passing electrons to protons (H^+^) to form H_2_. H_2_ in the hydrogenosome can also be produced in a determined way by hydrogen dehydrogenase; however, the enzyme mainly operates the reverse reaction to converse NAD(P)^+^ and H_2_ to H^+^ and NAD(P)H, which is in an energetically favorable direction. In addition, a speculative way of bifurcating hydrogenase to produce H_2_ has been put forward and seems important in the coupling of the reduction of H^+^ to the oxidation of NAD(P)H through the ferredoxin to generate H_2_ [41]. ATP is formed by substrate-level phosphorylation [90,91] (Figure 5).

#### 3.3.2. Amino Acid Metabolism

Carbohydrates are the first source of energy for a living organism. Living organisms will absorb other nutrients, such as amino acids, to sustain its growth when carbohydrates are insufficient. Previous studies have shown that under normal culture conditions, *T. vaginalis* consumes large amounts of arginine and small amounts of methionine for energy production; and when maltose is deficient, it consumes higher amounts of amino acids, especially arginine, threonine, and leucine [92].

Mukherjee et al. reported the identification and analyses of two glycine cleavage H proteins and a dihydrolipoamide dehydrogenase (L protein) from *T**. vaginalis*, which are pivotal members of the glycine decarboxylase complex, which catalyzes the oxidative decarboxylation and deamination of glycine. They determined their location in the hydrogenosome by immunofluorescence analysis, indicating the important role of the hydrogenosome in amino acid metabolism [93]. The amino acid synthesis and decomposition are associated with the Krebs cycle, glycolytic pathway, and pentose phosphate pathway through conversion to intermediate products (such as pyruvate, acetyl CoA, oxaloacetic acid, acetoacetic acid) in the glucose metabolism in the mitochondria. Similarly, metabolism of amino acids is linked with glycolysis for ATP production in the hydrogenosome. Many enzymes associated with amino acid metabolism and other related metabolic pathways in the protein metabolism at the gene level have been identified [94]. In addition to using immunology at the protein level, or RNA sequencing at the transcriptional level to analyze the differential expression of genes under changing environmental conditions, Huang et al. used LC–FTMS for the first time to analyze the regulation of amino acid metabolism in the hydrogenosome of *T. vaginalis* and its adaptive response in a glucose restriction environment at the metabolome level. The results showed that amino acid metabolism in the hydrogenosome of *T. vaginalis* was significantly altered in the case of carbohydrate deficiency. The endergonic metabolism of methionine was less stimulated to reduce ATP consumption, while the branched chain amino acid synthesis related to the glutamate metabolic pathway was markedly enhanced for energy generation [95].

#### 3.3.3. The Unique Metabolism of the Hydrogenosome in Anaerobic Fungi

The findings on the metabolism of the hydrogenosome are mainly based on the results of experiments conducted in the past several decades using trichomonas as a model. Therefore, the conclusions cannot prove that all hydrogenosomes have the same enzymes and metabolic activities. In fact, the enzyme composition and material metabolism in the hydrogenosome of anaerobic fungi are different from that of other microorganisms (Figure 5). Unlike other anaerobic microorganisms, anaerobic fungi depend on pyruvate: formate lyase (PFL), which exists in both the cytoplasm and the hydrogenosome to hydrolyze pyruvate into isomole formate and acetyl CoA, rather than PFO. Though many studies have suggested that PFO is either absent or of only marginal importance in the anaerobic fungal hydrogenosomal metabolism, both enzymes were identified in all published gut fungal genomes, and a metabolic model proposed and analyzed by Wilken et al. suggested that PFL carries more metabolism flux than PFO in the hydrogenosome, but an energetically favorable route to H_2_ production still requires the action of PFO [41]. The pyruvate in the cytoplasm is eventually metabolized to ethanol, and the pyruvate that enters in the hydrogenosome is eventually metabolized to acetate. Pyruvate can be directly transported into the hydrogenosome, and also may be formed from the dehydrogenation of malate entered into the hydrogenosome. CO_2_ and NAD(P)H are also produced during the dehydrogenation of malate, and then hydrogenase reduces 2H^+^ to H_2_ with or without the reducing equivalent of NAD(P)H. Acetyl-CoA formed in the hydrolysis of pyruvate first generates acetate and succinyl-CoA under the action of succinyl-CoA transferase. Then succinyl-CoA is converted to succinate by succinyl-CoA synthase, along with the release of CoA and conversion of ADP to ATP, which supplies energy for anaerobic fungi. Therefore, PFL is one of the most important enzymes in the sugar metabolism of anaerobic fungi. Thus, in addition to H_2_, CO_2_, acetate, lactate, ethanol, and succinate, the fermentation products of carbohydrates in anaerobic fungi contain formate in equal molar quantities with acetate plus ethanol [96].

So far, there have been no reports on the metabolism of amino acids in the hydrogenosomes of anaerobic fungi.

## 4. The Action of the Hydrogenosome Involved in Promoting the Utilization of Biomass

The hydrogenosome maintains an internal redox equilibrium by releasing H_2_, which is a continuous process. Some microbes can use H_2_ to produce other organic moieties. This interspecific interaction not only strengthens the ability of anaerobic fungi to degrade biomass through removing the inhibition to the hydrogenosome [97], but also converts H_2_ and CO_2_ (not used by anaerobic fungi) into other substances that can be of value, greatly improving the agricultural and industrial utilization worth of biomass (Figure 6).

### 4.1. The Role of the Hydrogenosome towards CH_4_ Generation

CH_4_ is a sustainable, clean, and renewable energy source [98]. CH_4_ extracted in nature is exhaustible, so it must be supplemented artificially. The lignocellulosic wastes, such as agricultural and forestry residues, are considered promising feedstocks for the production of CH_4_ because of their abundant availability in nature [99,100]. Their rational use can not only solve the waste disposal problem, but also increase the availability of renewable energy. The anaerobic fermentation of agricultural wastes to produce CH_4_ by coculture of anaerobic fungi and methanogens has been studied [101,102,103]. Anaerobic fungi in anaerobic fermentation systems provide sufficient growth substrates and methanogenic substrates, such as H_2_, formate, and acetate, for methanogens. Among them, H_2_ is the main CH_4_-producing substrate because of the advantages hydrotrophic methanogens offer. It is reported that the redox reaction of H_2_ and CO_2_ forms 82% of CH_4_ in the rumen [104]. The H_2_ produced by the hydrogenosome is consumed through interspecific H_2_ transfer (Figure 6), and the process is necessary for anaerobic fungi, because this maintains coupling of the H^+^ reduction and the oxidation of NAD(P)H, resulting in the regeneration of NAD(P)^+^, which is indispensable for maintaining redox balance in the cell. The activity of methanogens can stimulate the activity of anaerobic fungi and promote the metabolism of the hydrogenosome [9,105]. When they are cocultured, the carbon flux metabolized in the anaerobic fungal hydrogenosomes increases from 41.7% to 58.8% compared with that in the pure culture [10], resulting in higher production of the hydrogenosomal metabolites and consequent CH_4_. Therefore, this explains, to a certain extent, how (a) the production and utilization of H_2_ in the hydrogenosome links these two organisms together and (b) the regulation of the hydrogenosome metabolism could reduce or increase CH_4_ production.

### 4.2. The Role of the Hydrogenosome towards Acetate Generation

Acetate, an important carboxylic acid, is widely used in paint, plastics, adhesives, and other manufacturing industries and foods [106]. The global annual demand for acetate is estimated to be about 10 million tonnes [107]. Although acetate is mainly synthesized chemically, it can also be produced by anaerobic fermentation of biomass [107,108]. In previous studies, poplar branches were used as raw materials for mass production of acetate. However, in these experiments, biomass was converted into soluble sugars through pretreatment and enzymatic hydrolysis, and then acetic acid-producing bacteria were added for the fermentation [106,109]. In practice, some anaerobic homoacetogens can use H_2_ and CO_2_ to produce large amounts of acetic acid, which makes them attractive for using gas mixtures as a cheap and abundant source of carbon and energy [106,107]. Ni et al. investigated a bioreactor that coupled anaerobic fermentation and homoacetogenesis. Experimental results demonstrated that the H_2_ produced from anaerobic glucose degradation could be used immediately to produce acetate by homoacetogens with a relatively high acetate yield, and the process could be further enhanced with an introduction of the fed-batch operation mode [110]. The results implied the feasibility of acetate production by coculture of H_2_-producing anaerobic fungi and H_2_-utilizing homoacetogens (Figure 6). In order to avoid the adverse effects of high concentrations of acetate on bacteria and to recover high-pure acetate, efficient and cost-competitive membrane filtration technology [111], distillation, and extractant recovery [112] can be applied to improved fermentation equipment.

### 4.3. The Potential Role of the Hydrogenosome in Improving the Nutritional Value of Feed

The coculture of anaerobic fungi and H_2_-utilizing bacteria is expected to produce some beneficial nutrients (such as NH_3_-N and propionate) (Figure 6) to improve the nutritional value of the feed. NH_3_-N is important for protein supply for ruminants. It can be absorbed by rumen microbes [113,114] and converted into microbial proteins that can be used by the host. Many microbes, such as *Selenomonas ruminantium*, *Veillonella parvula*, and *Wolinella succinogenes* [115,116], in the rumen can generate NH_3_-N using H_2_ and formate when nitrate and nitrite are used as electron acceptors [117,118]. Anaerobic fungi are a recognized degrader of forages [119,120], and their efficiency depends on the utilization of H_2_ produced by the hydrogenosome. Using a symbiotic microbial population consisting of anaerobic fungi and these NH_3_-N-producing bacteria in the pretreatment of roughage may be a potential method to improve the digestibility and feeding value of feedstuffs. Although some studies opine that the rate of nitrite generation is much faster than the rate of conversion of nitrite to amine, and the accumulation of nitrite will cause adverse effects on other microorganisms and even the host body [121,122], the effect seems to be alleviated by adding more nitrate- and nitrite-reducing bacteria (increasing the number of nitrate-using microorganisms) [123], giving animals an adaptation period (increasing the ability of microorganisms to adapt to and use nitrates) [117,124], or encapsulating nitrate (slowing down the formation of nitrite) [125,126].

Propionate is an important gluconeogenesis substrate for ruminants [127,128]. Fumarate-reducing bacteria are hydrogenotrophic microorganisms that can use fumarate as H_2_ sink to produce propionate [129,130]. For every mole of propionate produced from fumarate, a stoichiometrically equal amount of H_2_ is consumed [131]. Thriving anaerobic fungi continuously produce H_2_ through their hydrogenosome, an essential substrate for fumarate-reducing bacteria to produce propionate. Thus, the coculture of anaerobic fungi and fumarate-reducing bacteria could be applied for feed pretreatment using fumarate as an additive. Although many studies have reported that fumarate can increase propionate proportion [132,133,134], the effect of the sole addition of fumarate is limited, and the current cost of fumarate is high, which makes its addition uneconomical [129]. Therefore, fumarate-reducing bacteria can be used as biotic agents to intensify fumarate conversion to propionate. Fumarate-reducing bacteria are common in the rumen [135], and some strains have been isolated [136]. Such bacteria have high fumarate reductase activity during fermentation [129,137]. These findings suggest that the interactions between these H_2_-utilizing microbes and H_2_-producing anaerobic fungi may be a viable way to prepare higher-quality ruminant feeds.

### 4.4. Expansibility and Challenges

In Section 4, interactions between cocultured anaerobic fungi and H_2_-utilizing microorganisms that can promote the utilization of lignocellulosic biomass owing to the activity of the hydrogenosome were theoretically proposed. These microorganisms use H_2_ to produce other value-added products, and the utilization of H_2_ leads to the reduction of the H_2_ partial pressure in the system and removal of inhibition on the hydrogenosome, so as to catalyze more NAD(P)H to produce H_2_ [10]. The process also gives full play to the strong degradation ability of anaerobic fungi to crude fiber. Physical and chemical pretreatment has been widely used in the utilization of lignocellulosic biomass for a long time, but these technologies inevitably bring some cost and environmental problems. In recent years, there is a considerable interest in developing more sustainable, biobased pathways for lignin deconstruction. Anaerobic fungi with strong fiber-degrading ability are expected to provide promising insights for that. Compared with the cumbersome physical and chemical degradation of biomass, anaerobic fungi are expected to complete the pretreatment of lignocellulose in a more efficient and environmentally friendly way. The evaluation of its industrial value has been reported in many articles [43,138,139]. The degradation of lignocellulose by anaerobic fungi is comparable to that by steam explosion (physical treatment) [138] and by enzymatic hydrolysis (chemical treatment) [42]. However, there is no doubt that mild chemical pretreatment seems to be further beneficial to the transformation of biomass into biological products by anaerobic fungi, which was demonstrated by Youssef et al. [42]. Therefore, it can be considered that the mutualistic coculture of anaerobic fungi and other microorganisms can promote the degradation of lignocellulose by anaerobic fungi and the production of various by-products, and moderate physical and chemical pretreatment can achieve a more efficient and ideal output.

Additionally, although H_2_ transfer between bacteria has also been observed, the anaerobic fungi which have more advantageous features (physical invasion to plant tissue and higher abundance of highly specialized CAZymes) than bacteria were selected to coculture with H_2_-utilizing microorganisms for further discussion. The advantage of anaerobic fungi is also proved by experiments. For example, an in vitro experiment by Ma et al. showed that the digestibility of lignocellulosic substrates and methane yield in the enrichment of anaerobic fungi and methanogens were significantly higher than the enrichment of bacteria and methanogens [139]. However, it is worth noting that the practical application of these proposed coculture systems is still facing many challenges, such as maintenance of an anaerobic, sterile environment in a bioreactor, instability of an anaerobic fungal in vitro culture system, comparison between microbial coculture and physical and chemical treatments, and slower degradation rate of roughage by microbial preparation. In fact, although it is true that bacteria can multiply faster than fungi, there are many experiments showing that coculture of bacteria and fungi can actually develop for a long time and be of great activity [140,141]. Further, mutualistic relationships have been probed to help set up stable eukaryote–bacteria consortia used for anaerobic fermentation [142,143,144]. These reports showed that the stable microbial community structure was related to the cocultured bacterial types [144] and the inoculation proportion of eukaryotes–bacteria [142,143]. As for the feasibility of using anaerobic fungi for in vitro fermentation, there were previous experiments that directly inoculated anaerobic fungi on lignocellulosic biomass as silage additive for long-term monitoring, and achieved positive effects [145,146]. Podolsky et al. also proposed that the appropriate dense and volume of the inoculated culture, and optimal inoculation methods and bioreactor conditions should be paid attention to in the practical application of anaerobic fungi [147]. However, most of the current research studies are still limited to small-scale cultivation in the laboratory. It does require necessary cost and time to test the feasibility of large-scale in vitro cultivation and inoculation because of the harsh anaerobic and sterile environment required by anaerobic fungi.

All in all, this challenging and meaningful undertaking needs more trials to perfect.

## 5. Conclusions and Expectation

This review summarized the classification and biomass degradation characteristics of anaerobic fungi. In addition, the metabolic characteristics of anaerobic fungi’s special metabolic organ—the hydrogenosome—and its role in biomass utilization to produce valued bioproducts were discussed. With the development of molecular tools, more and more information about the metabolism of the hydrogenosome (a vital organ that has long been neglected) is being obtained. The metabolism of the hydrogenosome is integral to the function of anaerobic fungi, and its metabolites are valued substrates for other microorganisms. With further research, we believe that the generation and utilization of H_2_ in the hydrogenosome have great potential to enhance the degradation of biomass and generate value-added products. More experiments that focus on metabolic regulation in the hydrogenosome should be conducted so that the hydrogenosome can work more prominently in a microbial consortium to obtain valuable outputs. This will bring considerable economic and environmental benefits.

## Figures and Tables

**Figure 1 jof-08-00338-f001:**
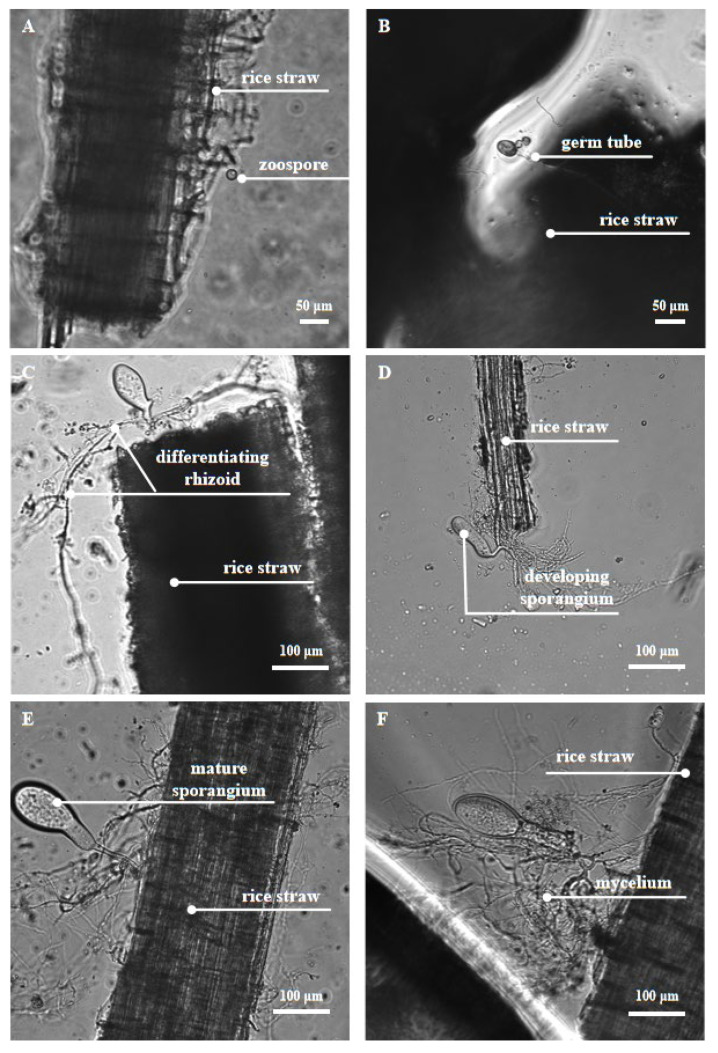
Anaerobic fungus *Pecoramyces* sp. F1 under the phase contrast microscope. The anaerobic fungus *Pecoramyces* sp. F1 undergoes the growth stages of zoospore attachment (**A**), rhizoid growth and penetration into the rice straw surface (**B**–**F**), and sporangium development and maturation (**C**–**E**).

**Figure 2 jof-08-00338-f002:**
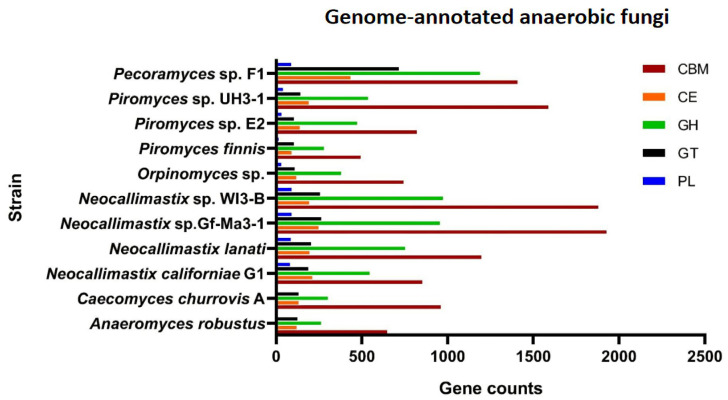
Genome-annotated information of anaerobic fungi. Anaerobic fungi strains that have been genome-annotated (*Caecomyces churrovis* A, *Anaeromyces robustus*, *Neocallimastix californiae* G1, *Neocallimastix lanati*, *Neocallimastix* sp. Gf-Ma3-1, *Neocallimastix* sp. WI3-B, *Orpinomyces* sp., *Piromyces finnis*, *Piromyces* sp. E2, *Piromyces* sp. UH3-1) can be found in JGI’s MycoCosm (https://mycocosm.jgi.doe.gov/mycocosm/annotations/browser/cazy/summary;pEimlQ?p=neocallimastigomycetes (accessed on 10 December 2021)), and the results of their annotation in the CAZymes database are included. The genome of *Pecoramyces* sp. F1 was uploaded to dbCAN2 [49] for online annotation, and at the same time, BLAST [50] was used for the annotation of gene models against CAZymes database too. The data of AAs presented in the annotation results of *Pecoramyces* sp. F1 but that of other 10 strains cannot be found in JGI’s MycoCosm (https://mycocosm.jgi.doe.gov/mycocosm/annotations/browser/cazy/summary;pEimlQ?p=neocallimastigomycetes (accessed on 10 December 2021)), so the data of AAs are not shown. dbCAN2 and BLAST results were combined to get the *Pecoramyces* sp. F1 genome annotation results in the figure.

**Figure 3 jof-08-00338-f003:**
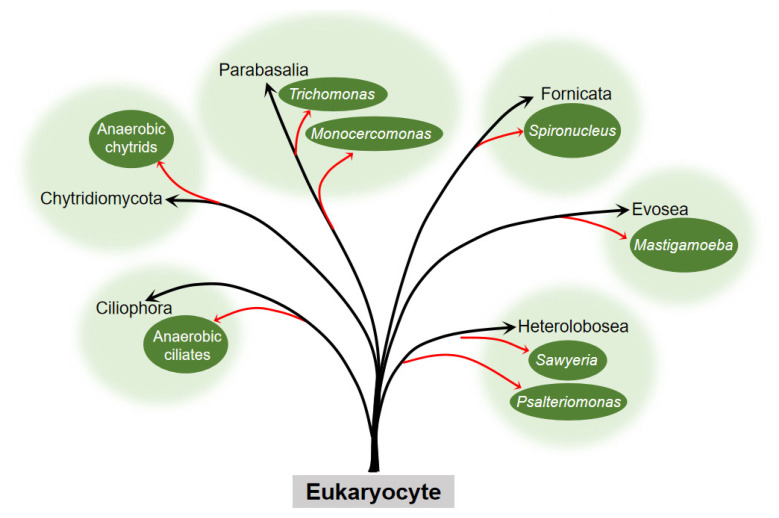
Occurrence of the hydrogenosome within the six eukaryocyte supergroups. Embranchment reflects the taxons of eukaryocyte with the hydrogenosome. The black arrow points to the phylum, and the red arrow points to the clades or genus to which the eukaryocyte belongs. Branch length is not to scale.

**Figure 4 jof-08-00338-f004:**
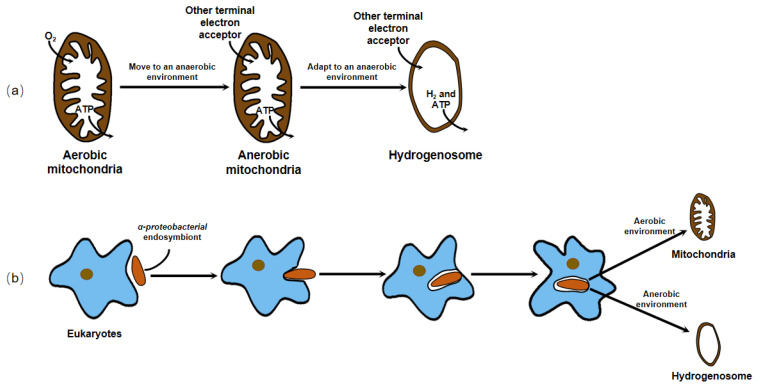
There are two dominant hypotheses about the origin of the hydrogenosome. (**a**) The hydrogenosome is a degraded form of mitochondria that lost some functional proteins and gene fragments to adapt to an anaerobic environment. (**b**) The hydrogenosome and mitochondria originate from the same or different endosymbionts to cope with different atmospheric conditions and energy driving forces.

**Figure 5 jof-08-00338-f005:**
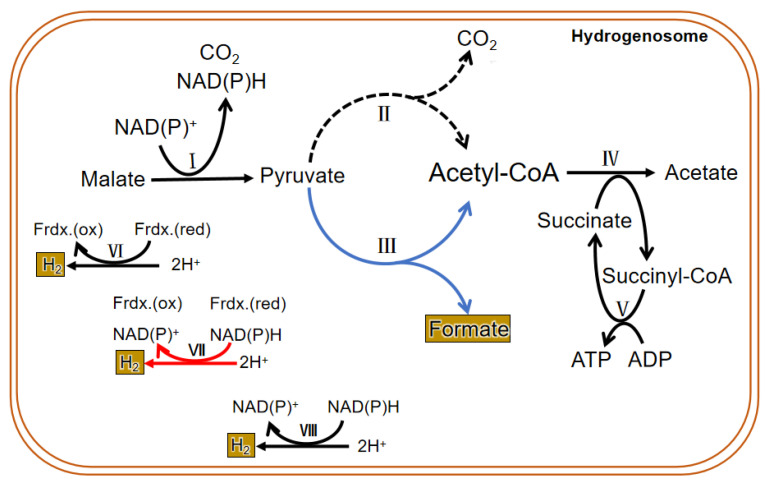
Metabolic process in the hydrogenosome of both trichomonas and anaerobic fungi. The dotted line is the main metabolic pathway of trichomonas hydrogenosomes. The blue line represents the main hydrogenosomal metabolism pathway of anaerobic fungi. The black line is the common metabolic pathway of both of them. The red line shows a suspected H_2_ generation pathway, and it commonly exits in both. I. malic enzyme; II. pyruvate: ferredoxin oxidoreductase (PFO); III. pyruvate formate lyase (PFL); IV. acetate: succinyl CoA-transferase; V. succinyl-CoA synthetase; VI. ferredoxin hydrogenase; VII. bifurcating hydrogenase; VIII. hydrogen dehydrogenase.

**Figure 6 jof-08-00338-f006:**
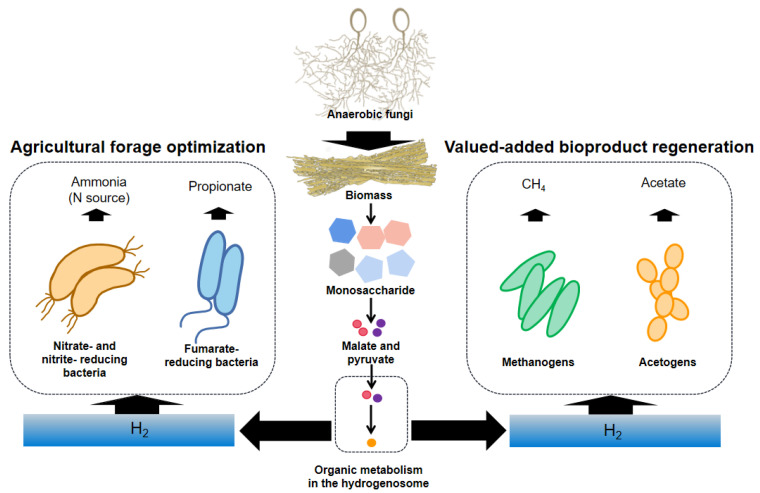
The role of the hydrogenosome and its metabolites. The metabolites (mainly H_2_) can be used by nitrate- and nitrite-utilizing bacteria, fumarate-reducing bacteria, methanogens, and acetogens to produce other substances that can be used.

**Table 1 jof-08-00338-t001:** The anaerobic fungal isolates (11) that have been genome-annotated in recent years.

Organism	Assembly Length	Genes Count	Isolation Source	Sample	Published
*Piromyces* sp. UH3-1	84,096,456	16,867	Donkey	Feces	-
*Piromyces* sp. E2	71,019,055	14,648	Elephant	Feces	[39]
*Piromyces finnis*	56,455,805	10,992	Horse	Feces	[39]
*Caecomyces churrovis* A	165,495,782	15,009	Sheep	Feces	[40]
*Anaeromyces robustus*	71,685,009	12,832	Sheep	Feces	[39]
*Neocallimastix* sp. Gf-Ma3-1	209,503,801	28,646	Giraffe	Feces	-
*Neocallimastix* sp. WI3-B	206,810,295	28,960	Wildebeest	Feces	-
*Neocallimastix lanati*	200,974,851	27,677	Sheep	Feces	[41]
*Neocallimastix californiae* G1	193,032,486	20,219	Goat	Feces	[39]
*Pecoramyces ruminantium* C1A	100,954,185	18,936	Angus steer	Feces	[42]
*Pecoramyces* sp. F1	106,834,627	17,740	Goat	Rumen sample	[43]

*Piromyces* sp. UH3-1, *Neocallimastix* sp. Gf-Ma3-1 and *Neocallimastix* sp. WI3-B are three strains recorded in JGI’s MycoCosm (https://mycocosm.jgi.doe.gov/mycocosm/species-tree/tree;ltOh_E?organism=neocallimastigomycetes (accessed on 10 December 2021)), but no references were found. All genome data can be downloaded at JGI (https://genome.jgi.doe.gov/portal/neocallimastigomycetes/neocallimastigomycetes.download.html (accessed on 10 December 2021)).

## Data Availability

Not applicable.

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
