# Peer review of "Hydrogenosome, Pairing Anaerobic Fungi and H2-Utilizing Microorganisms Based on Metabolic Ties to Facilitate Biomass Utilization"

_jof, 2022, doi:10.3390/jof8040338_

Round 1

Reviewer 1 Report

The first portion of the review (Sections 1 through 3) provide a useful, nicely done introduction to hydrogenosomes, including the characteristics unique to the fungal hydrogenosomes, which have received less study than have those of the anaerobic ciliates or Trichomonas. The reviewer is much less enthused about Section 4, in which the authors propose coupling biomass degradation by the anaerobic fungi with various hydrogen-consuming procaryotes for the production of ammonia, sulfide, propionate, methane or acetate. In the reviewer’s opinion, these sections describe coculture systems that either have, or probably could be, demonstrated in the laboratory, but would be woefully impractical in an industrial or agricultural setting, that represents the focus of the authors’ interest.

            As the reviewer interprets section 4.1, it appears that the authors are trying to make a case for pretreating feeds with cocultures of H2-producing fungi (which partially degrade plant biomass) and H2-consuming bacteria that will assist the process by thermodynamic displacement (viz., removing H2 to produce ammonia, sulfide or propionate). The problem here is that both plant biomass degradation and formation of the above products already occur in the rumen through normal metabolism of the native microbiota. Additionally, the products are very low in value (sulfide, ammonia) or are at least lower than the value of the proposed substrates (as in the case of propionate from fumarate). In terms of producing these compounds outside the rumen (in a bioreactor), all of them can likely be produced by combining these same procaryotes with H2-producing bacteria that can be grown more easily and more rapidly than the anaerobic fungi. Additionally, the fungal/procaryote cocultures would be difficult to maintain as defined cocultures without sterilization of feedstock, equipment, etc. – an expensive proposition. If one wanted to produce these compounds in a bioreactor, it would seem to make far more sense, practically and economically, to conduct fermentations with “open cultures”, i.e., stable mixed communities that can be maintained without sterilization requirements. But such cultures would soon select out the fungi due to the latter’s relatively slow growth (just as fungi disappear rapidly from conventional in vitro ruminal fermentations).

            In the reviewer’s opinion, the schemes proposed in Section 4.2, for production of methane or acetic acid from lignocellulose using cocultures of ruminal fungi and methanogens or acetogens, respectively, are also highly impractical. Certainly, one can establish co-cultures in the laboratory that will perform the desired transformations, but again, unless expensive sterilization was employed, these cocultures could not be maintained in an industrial setting. In the case of the acetogens, they are rapidly outcompeted by methanogens, and keeping the latter group at bay would probably require addition of toxic inhibitors of methanogenesis – not a practical industrial strategy if waste disposal is consideration. The authors neglect to discuss (or even mention), as noted above, for all their capacity to degrade plant biomass, ruminal fungi lack competitiveness outside the rumen. The authors do not at all consider this in the naïvely constructed scenarios proposed in this review.

Specific comments:

L18: Here and elsewhere in the manuscript: The hydrogensome is not an “organ”. It is an organelle.  

L38-39: This is a gross misreading of the cited reference [3]. In that reference, Akin et al focus only on the rumen, and make no statement regarding the percentage of roughage degradation attributable to ruminal fungi. What they do state (in their abstract) is that, for fungal monocultures isolated from the rumen, “In in vitro digestion studies over a period of 9 days, all isolates eventually degraded ~ 70% of the leaf dry matter”, and about 45% of stem dry matter This is not at all the same thing – degradation (expressed on the basis of percentage of added biomass) can always be increased by adding smaller amounts of biomass. Most researchers would ascribe an important role for fungi in physical disruption of plant tissue, but overall a minor overall role in plant biomass degradation in the rumen.

L53-61: This paragraph needs to be rewritten more concisely.

L118: Do the authors mean a greater relative change (as opposed to “a relative change”)?

L125-127: This is a little confusing. If the LSU sequences are more conserved and demonstrate less heterogeneity, how does this improve, rather than diminish, their use in phylogenetic characterization?

L138: It would be useful to include the range of AT percentages here.

L148: Not sure what the authors mean here. Higher concentrations of which metabolites? Higher concentration than what? What is “chemical digestion” in this, an enzymatic process?

L162: Are the enzymes releasing cellulose and hemicellulose, or are they releasing oligomers of these polysaccharides? This is an important distinction: Releasing the bulk polymers implies an enzymatic cleavage of lignin-carbohydrate bonds, while releasing oligomers only requires the breaking of glycosidic bonds within the lignocellulosic matrix.

L171: “400-fold’ magnification is a meaningless term here, as the magnification depends on the size of the digital image, which can be enlarged and reduced to any level. The authors should instead insert scale bars in each panel indicating, for example, 1 or 10 micrometers.

L210: What is meant by “survival substrates”?

L221: Oligomers are also products, sometimes the main products, of some of these enzymes.

L264: What is meant by “taxonomically irrelevant”? In the biological world, no organism is “irrelevant”!

L277-278: Sentence needs rewriting. Classical endosymbiotic theory posits that the cells that engulfed the aerobic bacteria (the precursors to mitochondria) were other bacteria, not yet eucaryotes.

L333: How close?

L390-391: By “the forward, energetically favorable direction”, do the authors mean H2 oxidation and NADP+ reduction?

L316: Associated in what way?

L428: If the goal is reduced ATP consumption, why were amino acid biosynthetic reactions (which require ATP) activated?

L452: What body?

Figure 6: The authors have incorrectly switched the products of the nitrate/nitrite-reducing and sulfate-reducing bacteria.

L510-513: Sulfur is required by ruminants, but it is rarely limiting for cysteine synthesis. The critical need for sulfur is for methionine, an essential amino acid which is typically supplied in the feed, often in a rumen-protected form.  Providing inorganic sulfur will not meet this need.

L514-516: The effect of sulfide in the in vitro system described could simply have been due to the ability of the sulfide to better establish the reducing conditions need for successful fermentation by the anaerobic microbial community.

L517-520: The authors do not delve into this sulfide toxicity problem. Indeed, excess sulfide production in the rumen can result in polioencephalomalacia, which can cause not merely discomfort, but death of the animal; this is why sulfate levels in animal rations are typically very low. Excess sulfide is also problematic in that it readily binds essential trace metals such as copper, decreasing their availability.

L569: By “mainly”, the authors seem to imply that anaerobic fungi have been studied as agents for conversion of ag wastes to methane to a greater extent than have bacteria. In fact, there is a long history of bacterial fermentations of ag wastes to methane, much more extensive than work with anaerobic fungi.

L572: Methanogens do not use ethanol as a substrate.

L575: This is likely true, but most of this H2 is produced by bacteria, not by fungi.

L576-585: This description for methanogenesis from fungal H2 can be equally applied to H2 produced by bacteria; in fact the entire concept of interspecies hydrogen transfer was demonstrated first using bacterial cultures.

L603-605: The mechanisms and energetics are, in fact, well-understood. What is less apparent is how one could make the proposed process economically feasible, owing to a host of challenges that include competition by methanogens, high cost of recovery of acetate from dilute aqueous solution, etc.

Minor edits:

L25: Change “a model strain” to “model organisms”.

L43: Subscript “2” in “H2”.

L97: Change “depends” to “depended”.

L98: Change “flagellates” to “flagella”.

L113: Change “conservative” to “conserved”.

L195: Change “extraglucanase” to “exoglucansase”.

L207, L235: Change “code” to “encode”.

L218: Change “prevents” to “prevent”.

L227: Superscript “-1”.

L232: Change “thermosi” to “thermocellum”, and “albicans” to “albus”.

L271: Change “Twenty hundred million” to “Two billion”.

L274: Change “has” to “had”.

L276: Change “transport receptor” to “acceptor”.

L298: Change “mitochondria” to “mitochondrion”.

L330: Change “fungi” to “fungus”.

L331: Delete “Marlene”.

L350: Subscript “2” in “CO2”.

L354: Define “PFO”.

L369, L406, L411: Italicize “T. vaginalis”.

L378: Change “ferredox” to “ferredoxin”.

L387: Change “oxidated” to “oxidized”.

L392: Change “forwarded” to “forward”.

Fig. 5: Change “Formte” to “Formate”.

L397, L442: Change “metabolism” to “metabolic”.

L399: Change ‘suspective” to “suspected”.

L401: Close parentheses after “PFL”.

L489: Italicize “S. ruminantium”.

L500: Change “opinion” to “opine”.

Reviewer 2 Report

The manuscript can benefit by addressing the following suggestions/comments:

  1. The manuscript is well-written. But, there are several misspelled words (e.g., relys in line 102), out of place words (e.g., "be" in line 152), and please also avoid first person sentences.
  2. The left box of Figure 6 has major errors that need to be fixed.
  3. The title of the manuscript is a little misleading. The title implied that the biomass utilization through anaerobic fungal hydrogenosome is better or an improvement of currently practiced route. However, the authors only laid out how anaerobic fungi could work with other microbial systems with no clear evidence of improvement.
  4. In my opinion, the most feasible route for converting (lignocellulosic) biomass to bioproducts is the one the involves chemical biomass pretreatment followed by biochemical process (e.g., anaerobic digestion using bacterial consortium that contains acidogens, acetogens, and methanogens to produce biogas). This is typically the baseline for comparison if any other process is proposed. Compared to this, how improved the process (e.g., acetate production, methane production) could be if fungal hydrogenosome is employed?
  5. The process proposed in the manuscript (fungal hydrogenosome to produce H2 without biomass pretreatment) could be a direct replacement to the bio-H2 production from chemically-pretreated biomass with simultaneous production of organic acids (acetate, propionate, butyrate, etc.) using acidogens/acetogens. Knowing that the initial fungal biomass degradation (of cellulose and hemicellulose) is going to be really slow, how can the proposed process be an improvement?
  6. When acidogens/acetogens are combined with methanogens (as is almost always the case for CH4 production), proton/NADH coupling or interspecies exchange have been observed as was discussed in the manuscript. What is/are the (dis)advantage/s of this to the coupling compared to the one that involves fungi?

Reviewer 3 Report

An interesting article on a relevant topic.

Please revise the English language of the manuscript, a few corrections are advised.

Page 2; Line 53: H2 that produced….omit ‘that’

Page 2; Line 61: ‘..and for gaining novel insights…’

Page 2; Line 65: ‘Since then, many researchers have..’

Page 3; Line 102: Correct ‘relys’ to ‘relies’

Page 5; line 146: ‘In comparison to bacteria…’

Page 5; Line 152: ‘It my provide..’

Page 6; Line 168: Italicise Pecoramyces

Page 8; Line 237: ..’while the latter…’

Page 11/12; Line 369:; 411 Italicise T. vaginalis

Page 14; Line 489: Italicise S. ruminantium

Round 2

Reviewer 1 Report

The authors have substantially revised the manuscript, and – to their credit -- have included caveats and qualifying statements brought up previously by the reviewer, particularly in section 4 of the manuscript. These concerns, while now presented to the reader, do provide a more balanced presentation. But this balanced presentation makes most of the authors’ proposals seem even more impractical. Thus, the reviewer remains concerned regarding the whole thrust of the review. For example, in section 4.1, are the authors proposing the feedstuffs be fermented outside the rumen (i.e., in a bioreactor) by a defined coculture of ruminal fungi and a nitrate-reducing bacterium, as a means of pretreating the feedstuff and producing NPN? This simply does not seem economically feasible, because obtaining the desired effect would necessitate axenic culture conditions (sterilizing reactor and feedstuff) and adding costly nitrate, then recovering the products, all for the weak economic benefit of a slightly more fermentable feedstuff and some microbially available ammonia. In section 4.2, are the authors proposing that a defined mixed culture of SRB and ruminal fungi be used to produce H2S, for use as a feed additive? This makes even less sense: Not only do the above concerns regarding axenic culture remain, but additional problems are introduced in that H2S is poisonous to handle (more toxic than hydrogen cyanide) and thus impractical to add to a feed. Moreover, it renders feeds unpalatable due to its famously putrid, “rotten egg” odor. If targeted production of H2S to the rumen is desired, it would be best to do this by adding carefully titrated amounts of very inexpensive sulfate directly to the feed, as SRB can use a wide variety of electron donors (including VFA) that are widely available in the rumen, for sulfate reduction to sulfide; there is no need or benefit to involving the ruminal fungi in the process.

            In other words, the authors have identified several beneficial products that can be produced by fungal coculture with various procaryotes. All of these products share two features: they are of relatively low economic value, and they can be produced in other microbial systems that do not involve ruminal fungi or hydrogenosomes.

            In summary, the authors do an excellent job of describing the characteristics, evolution, and ecological role of hydrogenosomes and of the ruminal fungi. By itself, the review’s first three sections represent a useful contribution to the literature. But Section 4 remains misplaced and diversionary, and substantially weakens the overall manuscript.

Specific comments.

L96: More clarity needed here. The point is that monocentric fungi do not have multiple nuclei in the myeclia, rather than not having nuclei at all, correct?

L171: The reviewer still feels that “chemical digestion’ is a misnomer here. The process described is purely enzymatic, as opposed to authentic chemical digestion, which uses diffusible reactive chemicals (protons, alkali, oxidizing agents, etc.). Suggest just titling this section “Digestion by diverse plant fiber-degrading enzymes”.

L274: Do the authors mean α-proteobacteria?

L320: Not clear regarding loss of “some genes”. Do hydrogenosomes contain any DNA at all?

L388-390: The reviewer still has trouble with this. Under most conditions, the reduction of protons to H2 is thermodynamically constrained (the standard reduction potential at pH 7 is -0.414 V; H2 production is only “thermodynamically favorable if H2 is continually kept at low partial pressure.

L453: What body?

L549: Which microbes use propionate as a nutrient?

L551-560: Regardless of the presence of fumarate-reducing bacteria, or of augmenting a community with additional levels of these bacteria, the fact remains that the limitation to propionate production is the supply of fumarate, which, as the authors point out, is too expensive to supply directly.

Minor edits:

L95: Change “flagellates” to “flagella”.

L217: Change “These simple sugars” to “The component monosaccharides”.

L242: Insert “and the bacteria” ahead of “Clostridium” to insure that the bacteria are not confused with the anaerobic fungi.

L258: Change “commercial enzyme preparations” to “commercial preparations containing non-complexed enzymes”.

L343: Subscript the “0” and “1” in “F0F1”.

L354: Change “get” to “be”.

L449: Change “reduction” to “reducing”.

L492: Change “bacteria” to “species”.

Reviewer 2 Report

  1. There are still misspelled words.
  2. Please avoid writing in first person.
